# The Effect of Storage Conditions on the Content of Molecules in *Malus domestica* ‘Chopin’ cv. and Their In Vitro Antioxidant Activity

**DOI:** 10.3390/molecules27206979

**Published:** 2022-10-17

**Authors:** Alicja Ponder, Elvyra Jariené, Ewelina Hallmann

**Affiliations:** 1Department of Functional and Organic Food, Institute of Human Nutrition Sciences, Warsaw University of Life Sciences, Nowoursynowska 159c, 02-776 Warsaw, Poland; 2Department of Agrobiology and Food Sciences, Agriculture Academy, Vytautas Magnus University, Donelaicio 58, 44248 Kaunas, Lithuania; 3Agriculture Academy, Bioeconomy Research Institute, Vytautas Magnus University, K. Donelaičio 58, 44248 Kanuas, Lithuania

**Keywords:** apple, antioxidant activity, vitamin C, carotenoids, chlorophyll, polyphenols, HPLC

## Abstract

The apple fruit is one of the most widely cultivated temperate fruits and a rich source of bioactive compounds. Since a growing number of consumers are increasingly seeking safe and healthy food production, organic apple production provides this possibility. In this study, the content of bioactive compounds in organic apples depending on temperature and storage time were determined. ‘Chopin’ apples were stored for 7, 14 and 21 days at 5 °C and 20 °C. All analyses were performed using the HPLC method. The control samples of fresh apples contained the highest content of vitamin C (21.87 mg/100 g FW), L-ascorbic (11.50 mg/100 g FW), DHA (10.37 mg/100 g FW), total carotenoids (2.82 mg/100 g FW), β-carotene (0.21 mg/100 g FW) and lutein (2.41 mg/100 g FW). Samples stored at 20 °C had the highest content of total carotenoids and lutein, but samples stored at 5 °C contained the most β-carotene and zeaxanthin. Apples stored at 20 °C contained significantly more chlorophyll a (0.05 g/100 g FW). The samples stored at 5 °C contained the most total polyphenols. Samples stored for 14 days were characterized by the highest content of total flavonoids. Samples stored for 21 days were characterized by the highest content of epigallocatechin and quercetin-3-O-rutinoside, but the highest content of quercetin and kaempferol was found in control samples. The highest content of total flavonoids was found in apples stored at 5 °C.

## 1. Introduction

The apple (*Malus domestica* L. Borkh), with its wide diversity of climatic adaptation, has become the most widely cultivated tree fruit of the temperate zone and one of the most widely planted fruits in the world. Over the last years, researchers’ attention to this species has grown exponentially, as shown by the increasing number of scientific studies dealing not only with different aspects of apple cultivation and postharvest practices but also with fruit consumption [1,2,3,4,5,6,7,8].

Apples are edible fruits. Many other products are produced from this fruit: preserves, juices, jams, compotes, tea, wine, ciders and dried fruit. They are very important in human nutrition since they increase immunity, because they contain many antioxidant substances that are beneficial for humans and have a positive effect on stress resistance. It has been proven that apples are healthy and have many health benefits [2].

Many scientific studies have confirmed that apple consumption offers a reduced risk of cancer and cardiovascular disease. Therefore, in order to broaden the knowledge about the positive health-promoting properties of apples, it is necessary to characterize the content of related substances, such as polyphenolic compounds. The effects on human health of polyphenolic compounds are based on the results of research that has increased interest in the consumption of foods containing polyphenols [3,4]. Such health-promoting properties include antioxidant activities and the prevention of many diseases caused by oxidative stress [5]. Research on apples has shown that these fruits have strong antioxidant properties and inhibit the growth of cancer cells in vitro [6,7,8].

Growing consumer concerns about the negative health effects of synthetic pesticides commonly used in conventional agriculture have contributed to an increasing demand for food from alternative systems. One such system that has grown in popularity in recent decades is organic farming. In order to reduce the impact of production inputs on the environment and reduce the risk of exposure of consumers to pesticide residues in raw materials, organic farmers do not use synthetic pesticides [9]. Additionally, such opposing food production systems used in organic and conventional farming can affect the yield but also the metabolism of plants, leading to differences in the composition of raw materials. Recent scientific research has shown that replacing synthetic fertilizers with organic fertilizers in agricultural production affects the profile of secondary metabolites (including polyphenols) in plant tissues by changing protein expression. As reported in a recently published meta-analysis [10], not only the profile but also the content of plant polyphenols can vary significantly between organic and conventionally grown raw materials, with organic fruit and vegetables generally having higher concentrations [11]. Some cultivars are especially recommended for organic farming. There is a large group of local, traditional apple cultivars that are less popular than commercial ones.

Considering the above, research on the profiles of compounds with antioxidant properties present in apple fruit is important and necessary for food producers and consumers. In this study, we assessed the synthesis of secondary metabolites in apples. The aim of the research presented in this paper was therefore to analyze and compare the concentrations of carotenoids, chlorophylls and phenolic acids, flavonols, vitamin C and antioxidant activity in the local apple cultivar ‘Chopin’ harvested in a certified organic orchard located in Poland. The influence of storage time and temperature on the content of bioactive compounds was also investigated.

## 2. Results

### 2.1. Dry Weight and Vitamin C Content

The tested apple samples contained, on average, 11.07–12.31 g of dry matter in 100 g of fresh weight. Two-factor ANOVA showed a significant impact of the time of storage and the temperature of storage on the dry matter content in the fruits (Table 1). The highest content of dry matter was found in samples stored for 14 days (12.31 g/100 g FW) and at 5 degrees (12.39 g/100 g FW).

The apple fruits within this study were also tested for dehydroascorbic acid (DHA) and L-ascorbic acid (L-ASC) contents. The time of storage and the temperature of storage have been shown to be significant factors affecting the contents of vitamin C (Table 1). The average content of dehydroascorbic acid in the fruit was at a similar level as the content of L-ascorbic acid. The control samples of fresh apple contained the highest content of vitamin C (21.87 mg/100 g FW), L-ASC (11.50 mg/100 g FW) and DHA (10.37 mg/100 g FW).

### 2.2. The Content of Carotenoids

The contents of β-carotene, lutein, and zeaxanthin in apple fruits were analyzed within this study. The time of storage and the temperature of storage showed a significant effect on the concentration of the total carotenoids β-carotene and lutein in the samples (Table 2). However, time did not affect the zeaxanthin content. The control samples had the highest content of total carotenoids (2.82 mg/100 g FW), β-carotene (0.21 mg/100 g FW) and lutein (2.41 mg/100 g FW). Samples stored at 20 degrees had the highest content of total carotenoids and lutein, but samples stored at 5 degrees contained the most β-carotene and zeaxanthin.

### 2.3. Chlorophyll Content

The chlorophyll (chlorophyll a and chlorophyll b) contents in apple fruits were also analyzed and, similar to carotenoids, the time of storage affected them significantly (Table 3). The chlorophyll b content was higher than that of chlorophyll a on average in all apple samples. The temperature of storage showed a significant effect only on the concentration of chlorophyll a. Apples stored at 20 degrees contained significantly more of this compound (0.05 g/100 g FW).

### 2.4. The Antioxidant Activity and Polyphenol Content

The antioxidant activity and polyphenol content in apple samples are presented in Table 4. The samples of apples stored the longest, i.e., for 21 days, had by far the highest antioxidant activity, total polyphenol, total phenolic acid, gallic and chlorogenic contents. In contrast, the control samples contained by far the most caffeic acid and *p*-coumaric acid. The samples stored at a refrigeration temperature of 5 degrees contained the most total polyphenols, total phenolic acids, gallic, chlorogenic, *p*-coumaric and ferulic acids.

Table 5 shows the content of total flavonoids and individually identified flavonoids in the apples. Samples stored for 14 days were characterized by the highest content of total flavonoids and catechin, and samples stored for 21 days were characterized by the highest content of epigallocatechin and quercetin-3-O-rutinoside, but the highest content of quercetin and kaempferol was found in control samples. The highest content of total flavonoids, catechin, quercetin-3-O-rutinoside and luteolin was found in apple stored at 5 degrees.

## 3. Discussion

The apple is currently the most widely cultivated and one of the most commonly consumed temperate fruits. It is considered one of the most important fruit sources of antioxidants, such as phenolics and vitamin C, in the human diet [6]. The profiles of these biologically active compounds in fruits and vegetables are known to be impacted by genetics (species, cultivar), storage conditions, the environment and agricultural factors, including those related to cultivation practices [12].

Various factors such as genotype differences, climatic conditions during cultivation, agricultural practices, maturity of raw materials and harvesting methods and post-harvest procedures can affect the vitamin C content of apples. The higher the intensity of sunlight during the growing season, the higher the vitamin C content in the fruit. The use of synthetic nitrogen fertilizers reduces the content of vitamin C. The content of vitamin C in apples can be increased with less frequent irrigation. However, post-harvest temperature modification is the most important factor in maintaining vitamin C in apples [13].

The time of storage and the temperature of storage have been shown to be significant factors affecting the vitamin C content. The fresh apple sample (control) had the highest vitamin C content (21.87 mg/100 g), and its amount decreased with storage time (17.91 mg/100 g after 7 days of storage; 17.19 mg/100 g after 14 days of storage; and 11.77 mg/100 g after 21 days of storage). The research work conducted by Anebi et al. [14] investigates the variation in vitamin C in apples after prolonged storage, which was found to decrease the content of vitamin C. The vitamin C content of apples was found to be 27.3 mg/100 g and 7.29 mg/100 g after 1 week of storage. In another study [15], the ascorbic acid (L-ASC), dehydroascorbic acid (DHA) and total vitamin C contents in apple pulp were quantified. The L-ASC/DHA ratio in both the apple pulp and peel increased throughout fruit development, whereas the L-ASC/DHA balance always shifted towards the oxidized form during storage and shelf life, putatively reflecting an abiotic stress response. Importantly, at all times during apple fruit development and storage, the apple peel contained a higher level of vitamin C than the pulp, most likely because of its direct exposure to abiotic and biotic stresses.

A study conducted by Torres et al. [16] investigated the effect of cold storage on apples and antioxidant metabolism during browning of the skin after harvest of the ‘Fuji’ cultivar grown under strong lighting and elevated temperatures. The effects of different antioxidant systems in tissues exposed to different levels of sunlight and subjected to varying degrees of light damage during cold storage were investigated. The incidence of browning increased with sun exposure. Both shaded and exposed fruit peel without sunburn symptoms had the highest L-ASC content.

Apart from chlorophylls, apple also contains carotenoids. Their concentration depends mainly on the level of chlorophyll. The higher the concentration of chlorophyll in the leaves, the more carotenoids. Chlorophyll is related to the function of carotenoids, which are produced by plants mainly to protect chlorophyll from photooxidation. Carotenoids are synthesized by the general biosynthetic pathway in plant chloroplasts [17]. In this study, the effect of storage on the decrease in the content of carotenoids (from 2.82 mg/100 g to 1.65 mg/100 g) and chlorophylls (from 0.40 mg/100 g to 0.34 mg/100 g) was found. A study conducted by Vondráková et al. [18] investigated how storage affects antioxidant levels in apples. Apples showed a steady decrease in total carotenoids (sum of β-carotene, lutein, neoxanthin, violaxanthin, zeaxanthin and anteraxanthin) over the storage period. Fruits stored under controlled weather conditions showed significantly higher levels of carotenoids than apples stored under normal conditions. Although the storage conditions had an effect on the carotenoid content of apples, the ratio between the levels of individual carotenoids was not dependent on the storage conditions or cultivar.

The apple fruit samples analyzed in this study contained 26.12 mg/100 g FW to 96.84 mg/100 g FW of phenolic acids (mainly chlorogenic acid). Along with the increase in the storage period, the content of phenolic acids also increased. In the case of the influence of the storage temperature, apples stored at a lower temperature were characterized by a higher content of phenolic acids than apples stored at a normal temperature. Mikulic-Petkovsek et al. [19] underlined that phenolics in fruit may be influenced by many factors, such as the fruit type, cultivation conditions, environmental conditions, growing season, storage environment, time and, lastly, processing and preservation methods. However, in their study, several different phenolic compounds were quantified. The levels of all analyzed groups of phenolics were higher (although not always significant) in the organically grown apple fruit than in the leaves or apples from integrated production. Moreover, organic production affected the increase in the antioxidant activity of apple peel. In our study, the samples of apples stored the longest, i.e., for 21 days, had by far the highest antioxidant activity. Another study [18] found that stored apples showed a higher phenolic acid content than freshly harvested apples, whereas both freshly harvested and stored apples showed similar levels of glycosylated phenolic acid. The storage conditions had no appreciable effect on each group of phenolic acids (total content or concentrations of individual compounds). The aim of another paper [20] was to study the detailed chemical composition of apples of three varieties (Ionathan, Golden Delicious and Starkrimson) at the end of the storage period. Samples were taken from the Romanian market. Differences were found, particularly in relation to the polyphenol content, carotenoids and chlorophyll.

## 4. Materials and Methods

### 4.1. Plant Material

The analysies was carried out at the Warsaw University of Life Sciences (WULS, Poland). The apple cultivar ‘Chopin’ was harvested in the 2020 vegetation season from a certified organic orchard located in central Poland (Łódź region, Marchaty (51°47′8″ N 20°29′13″ E)). Samples of ≥3 kg from the orchard were harvested in three replicates and delivered to the laboratory of Warsaw University of Life Sciences. The samples were stored for 7, 14 and 21 days at 5 and 20 degrees Celsius. The fruit was split into pieces. Fresh sub-samples from each sample were used to determine the dry weight. The remaining material was lyophilized using a Labconco freeze dryer (−45 °C, 0.11 mBar), milled in an A-11 lab grinder and stored at −80 °C prior to analysis.

### 4.2. Chemicals

Acetone (HPLC grade) from Sigma–Aldrich (Poland, Warsaw); diethyl ether (HPLC grade) from Merck; hexane (HPLC grade) from Sigma–Aldrich (Poland, Warsaw); methanol (HPLC grade) from Sigma–Aldrich; carotenoid standards (HPLC grade 99.5–99.9% pure) from Sigma–Aldrich; β-carotene, lutein, zeaxanthin and magnesium carbonate (pure) from ChemPur; sodium carbonate (analytical grade) from ChemPur; sodium chloride (analytical grade) from Sigma–Aldrich; and sodium peroxide (analytical grade) from ChemPur were used, with phenolic standards (purity 99.0–99.9%) from Sigma–Aldrich, and L-ascorbic acid (L-Asc) and dehydroascorbic acid (DHA) standards with 99% purity (Sigma–Aldrich, Poznan, Poland).

### 4.3. Dry Matter Analysis

The content of dry matter (DW) was examined before lyophilization, as described by Isaac and Maalekuu [21]. Empty glass beakers were weighed, filled with fresh apple pieces and weighed one more time. Samples were dried at 105 °C and 1013 hPa for 48 h in an FP-25 W Farma Play (Poland). After 72 h, the samples were cooled to 21 °C and weighed. The content of dry matter was calculated (in g 100 g^−1^ fresh weight (FW)) based on the heaviness difference.

### 4.4. Vitamin C Analysis

The content of vitamin C in the apples was conducted by the HPLC method (high-performance liquid chromatography) (HPLC), as previously described by Kazimierczak et al. [22], using a Shimadzu HPLC-set (Shim-pol, Warsaw, Poland) with two LC-20AD pumps, a CMB-20A set controller, a SIL-20AC autosampler and an SPD-20AV visible light detector with spectrum identification. The 100 mg freeze-dried powder was extracted with 5% metaphosphoric acid. The samples were mixed by a vortex mixer, incubated in an ultrasonic bath (15 min, 20 °C) abd and centrifuged (6000 rpm, 10 min, 0 °C). An amount of 100 μL of supernatant was injected onto a Phenomenex Hydro 80-A RP column (250 × 4.6 mm). The analysis parameters were as follows: analysis time of 18 min, mobile phase of 50 mM phosphate buffer (pH 4.4) and 0.1 mM sodium acetate and detection wavelength of 255–260 nm. L-ascorbic acid (L-Asc) and dehydroascorbic acid (DHA) were identified based on Fluka and Sigma–Aldrich (Warsaw, Poland) standards with HPLC purity (99%).

### 4.5. Carotenoid and Chlorophyll Analysis

The content of carotenoid and chlorophyll were determined by HPLC [23] (Figure 1). A freeze-dried apple fruit sample (100 mg) was weighed into a laboratory plastic tube with acetone (HPLC purity) and 10 mg of MgCO_3_. Samples were extracted in cold sonic bath (10 min, 0 °C, 5.5 kHz). The next samples were centrifuged (6000 rpm, 10 min, 0 °C). Supernatant was collected and analyzed by HPLC. For the determination of carotenoids, a Shimadzu HPLC-set (Shim-pol, Warsaw, Poland) with two LC-20AD pumps, a CMB-20A set controller, a SIL-20AC autosampler, an SPD-20AV visible light detector with spectrum identification, a CTD-20A oven and a Max-RP 80A column (size: 4.6 × 250 mm) were used. The gradient mobile phase was prepared from a mixture of acetonitrile and methanol (phase A): 90:10 and methanol and ethyl acetate (68:32). The time flow rate was 1.0 mL min^−1^; the wavelength range was 445–480 nm. For qualitative identification of carotenoid compounds, an external standard in the form of β-carotene, lutein, zeaxanthin, chlorophyll a and chlorophyll b (Sigma–Aldrich, Warsaw, Poland) of 99.9% purity was used (Appendix A) [23].

### 4.6. Antioxidant Activity

#### ABTS Reagent Preparation

20 mL of distilled water was added to 0.0265 g of potassium persulfate (K_2_S_2_O_8_). Next, 5 mL of distilled water followed by 5 mL of a previously prepared aqueous solution of potassium persulfate was added to 0.0384 g of ABTS (2′2-azinebis-3-ethylbenzothiazolin-6-sulfonic acid) reagent. The solution was prepared a minimum of 12 h before the planned assay and stored in the dark.

### 4.7. Antioxidant Activity Measurement

An amount of 250 mL of the sample of freeze-dried apple material tested was weighed into a sterile Falcon plastic tube with a cap (50 mL), and 25 mL of distilled water was added. It was placed onto a vortex shaker (LP shaker Vortex, Labo Plus, Warsaw, Poland) for 60 s at 2000 rpm for complete mixing. Subsequently, the sample was incubated in a shaker incubator (IKA KS 4000 Control, IKA, Staufen im Breisgau, Germany) for 60 min (temperature 30 °C, 6× *g*). After incubation, the sample was again shaken on a vortex shaker for 60 s for complete mixing and then centrifuged (centrifuge, MPW-380 R, Warsaw, Poland) at 5 °C and 14,560× *g* for 20 min. After centrifugation, the supernatant was used for measurements. In 10 mL glass tubes, test extract solution, measured with a predetermined dilution scheme (0.5–1.5 mL), was then added to 3 mL of ABTS cationic solution in PBS. Absorbance measurements were taken exactly 6 min after incubation at room temperature. Absorbance was measured at a wavelength λ = 734 nm using a spectrophotometer (Helios γ, Thermo Scientific, Warsaw, Poland). The obtained measurements were then converted using a special formula including the dilution factor, and the final results were expressed as mmol of TE (Trolox equivalents per 100 g FW (fresh weight of apple)) [24].

### 4.8. Polyphenol Content Analysis

Analysis of the phenolic acids and flavonols content was performed using the HPLC method, as previously described by Kazimierczak et al. (2019) [22]. The analyses were carried out using Shimadzu equipment (USA Manufacturing Inc., Auburn, AL, USA, two LC-20AD pumps, CBM-20A controller, a CTD-20AC oven, SIL-20AC autosampler, UV/Vis SPD-20AV detector). An amount of 100 mg of freeze-dried apple powder was mixed with 5 mL of 80% MeOH, shaken on a Micro-Shaker 326 M (Poland) and incubated in an ultrasonic bath at 30 °C for 10 min. Next, the sample was centrifuged (3450× *g*, 12 min, 2 °C), supernatant was collected and 500 μL of the supernatant was injected onto the HPLC Synergi Fusion-RP 80i Phenomenex column (250 mm × 4.60 mm). The polyphenolic compounds were separated under gradient conditions with a flow rate of 1 mL min^−1^ by applying an aqueous solution of 10% (*v*/*v*) acetonitrile (phase A) and 55% (*v*/*v*) acetonitrile (phase B), both acidified by ortho-phosphoric acid to pH 3.0.). The analysis parameters were as follows. The time of the analysis was 38 min. The phases changed as follows: 1.00–22.99 min 95% phase A and 5% phase B, 23.00–27.99 min 50% phase A and 50% phase B, 28.00–35.99 min 80% phase A and 20% phase B, and 36.00–38.00 min 95% phase A and 5% phase B. The detection wavelength was 250 nm for flavonols and 370 nm for phenolic acids. The identification of individual phenolics was based on Sigma–Aldrich and Fluka external standards with HPLC purity (99.00–99.99%) (Figure 2 and Appendix A).

### 4.9. Statistical Analysis

Received results were statistically analyzed. Analysis of variance with two factors was performed with Tukey’s test (*p* = 0.05) using Statgraphics^®^ Centurion 15.2.11.0 (StatPoint Technologies, Inc., Warranton, VA, USA). In the conducted study, two factors were analyzed: time and temperature. A lack of statistically significant difference (*p* > 0.05) is described in the tables as N.S. Different letters within a row indicate statistically significant differences at the level *p* < 0.05. Principal Component Analysis (PCA) was carried out with the XLSTAT Software package (XLSTAT, 2020, New York, NY, USA).

## 5. Conclusions

It is important for consumers to have access to high-quality nutrition products, abundant in health-promoting antioxidants. Our study demonstrated variation in the content of selected groups of bioactive compounds between apple fruit stored for 7, 14 and 21 days at 5 and 20 degrees Celsius.

The time of storage and the temperature of storage have been shown to be significant factors affecting the vitamin C content. The average content of dehydroascorbic acid in the fruit was at a similar level as the content of L-ascorbic acid. The control samples of fresh apple contained the highest content of vitamin C. The time of storage and the temperature of storage showed a significant effect on the concentration of the total carotenoids β-carotene and lutein in the samples. However, time did not affect the zeaxanthin content. The control samples had the highest contents of total carotenoids, β-carotene and lutein. Samples stored at 20 degrees had the highest content of total carotenoids and lutein, but samples stored at 5 degrees contained the most β-carotene and zeaxanthin. Carotenoids were degraded during the storage of the apples. During storage, oxidation reactions take place, in particular of vitamin C and carotenoids. As they grows, storage time and temperature decrease the share of the yellow color and clarity of the product as carotenoids are degraded. Similar to carotenoids, the time of storage had a significant impact on chlorophylls. The chlorophyll b content was higher than that of chlorophyll a on average in all apple samples. The temperature of storage showed a significant effect only on the concentration of chlorophyll a. Apples stored at 20 degrees contained significantly more of this compound. The samples of apples stored the longest, i.e., for 21 days, had significantly the highest antioxidant activity. The main antioxidants present in apples are vitamin C and polyphenols. Although the antioxidant activity of apples increased with the storage period, the content of vitamin C decreased with time. Therefore, it seems that the increasing antioxidant activity was positively correlated with the increasing polyphenol content in the stored apples, because the samples of apples stored the longest, i.e., for 21 days, had by far the highest content of total polyphenol, total phenolic acid, gallic and chlorogenic contents. In contrast, the control samples contained by far the most caffeic acid and *p*-coumaric acid. The samples stored at a refrigeration temperature of 5 degrees contained the most total polyphenols, total phenolic acids, gallic acid, chlorogenic acid, *p*-coumaric acid and ferulic acid. Samples stored for 14 days were characterized by the highest content of total flavonoids and catechin, and samples stored for 21 days were characterized by the highest content of epigallocatechin and quercetin-3-O-rutinoside, but the highest content of quercetin and kaempferol was found in control samples. The highest contents of total flavonoids, catechin, quercetin-3-O-rutinoside and luteolin were found in apples stored at 5 degrees. An important finding is that an increase in the content of polyphenols and flavonoids was noticed during storage. It turns out that the storage conditions caused a reaction that resulted in the production of secondary metabolites, which are polyphenolic compounds. Polyphenols are the natural protective mechanism of the plant, so the plant synthesizes higher amounts of polyphenols to protect itself against external factors.

These results, which provide insight into the identification of the storage conditions of apples with the highest quality characteristics, could be of interest for producers and consumers. However, the identified trends should be further confirmed with attention given to the potential interactions between organic and conventional systems, as well as location-specific growing conditions, in order to validate the conclusions. These results provide important information regarding the chemical composition of apple varieties from the Polish market for both human direct consumption and industrial processing.

## Figures and Tables

**Figure 1 molecules-27-06979-f001:**
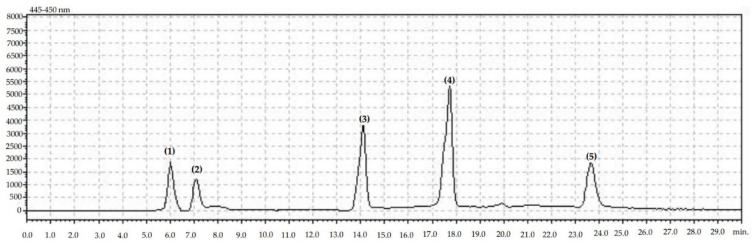
Chromatogram showing retention times for identified carotenoids and chlorophylles: (**1**) lutein, (**2**) zeaxanthin, (**3**) chlorphyll b, (**4**) chlorophyll a, (**5**) beta-carotene.

**Figure 2 molecules-27-06979-f002:**
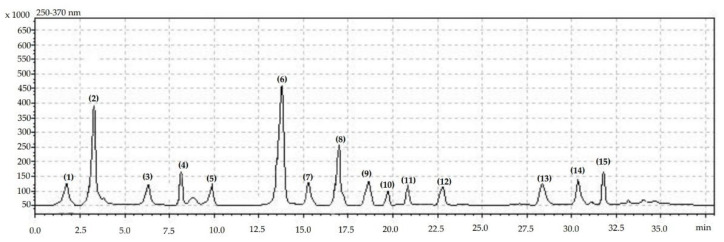
Chromatogram showing retention times for identified phenolic compounds: (**1**) phenols unknown, (**2**) gallic acid, (**3**) phenols unknown, (**4**) phenols unknown, (**5**) phenols unknown, (**6**) chlorogenic acid, (**7**) catechin, (**8**) epigallocatechin, (**9**) caffeic acid, (**10**) quercetin-3-O-rutinoside, (**11**) *p*-coumaric acid, (**12**) ferulic acid, (**13**) luteolin, (**14**) quercetin, (**15**) kaempferol.

**Table 1 molecules-27-06979-t001:** The content of dry weight (in g/100 g FW) and vitamin C in apple (in mg/100 g FW).

	Dry Weight	Vitamin C	L-Ascorbic Acid	Dehydroascorbic Acid
Time of storage				
Control	11.57 ± 0.05 ^cb^	21.87 ± 0.24 ^a^	11.50 ± 0.27 ^a^	10.37 ± 0.03 ^a^
After 7 days of storage	11.26 ± 0.13 ^c^	17.91 ± 0.21 ^b^	9.88 ± 0.22 ^b^	8.03 ± 0.02 ^b^
After 14 days of storage	12.31 ± 0.07 ^a^	17.19 ± 0.08 ^c^	9.38 ± 0.079 ^c^	7.81 ± 0.11 ^c^
After 21 days of storage	11.63 ± 0.38 ^ab^	11.77 ± 0.38 ^d^	5.72 ± 0.10 ^d^	6.05 ± 0.29 ^d^
Temperature of storage				
Storage at 5 degrees	12.39 ± 0.04 ^a^	15.01 ± 0.24 ^b^	7.49 ± 0.27 ^b^	7.52 ± 0.03 ^a^
Storage at 20 degrees	11.07 ± 0.04 ^b^	16.24 ± 0.21 ^a^	9.17 ± 0.08 ^a^	7.07 ± 0.20 ^b^
*p*-value				
Time (Tim)	0.001	<0.0001	<0.0001	<0.0001
Temperaturę (Tem)	0.0003	<0.0001	0.0496	<0.0001
Interaction Tim × Tem	0.0003	<0.0001	<0.0001	<0.0001

Data are presented as the mean ± SE with ANOVA *p*-value; means in columns followed by the same letter are not significantly different at the 5% level of probability (*p* < 0.05).

**Table 2 molecules-27-06979-t002:** The content of total carotenoids and individually identified carotenes in apple (in mg/100 g FW).

	Total Carotenoids	Lutein	Zeaxanthin	Beta Carotene
Time of storage				
Control	2.82 ± 0.04 ^a^	2.41 ± 0.04 ^a^	0.20 ± 0.00 ^a^	0.21 ± 0.00 ^a^
After 7 days of storage	2.33 ± 0.06 ^b^	1.93 ± 0.05 ^b^	0.20 ± 0.00 ^a^	0.20 ± 0.00 ^b^
After 14 days of storage	1.95 ± 0.03 ^b^	1.53 ± 0.04 ^bc^	0.21 ± 0.00 ^a^	0.21 ± 0.00 ^ab^
After 21 days of storage	1.65 ± 0.08 ^b^	1.27 ± 0.07 ^c^	0.19 ± 0.01 ^a^	0.19 ± 0.01 ^b^
Temperature of storage				
Storage at 5 degrees	1.58 ± 0.04 ^b^	1.16 ± 0.04 ^b^	0.21 ± 0.00 ^a^	0.21 ± 0.00 ^a^
Storage at 20 degrees	2.38 ± 0.08 ^a^	1.99 ± 0.08 ^a^	0.19 ± 0.00 ^b^	0.19 ± 0.00 ^b^
*p*-value				
Time (Tim)	<0.0001	<0.0001	0.3035	0.0026
Temperaturę (Tem)	<0.0001	<0.0001	0.004	0.0005
Interaction Tim × Tem	<0.0001	<0.0001	0.0003	0.0001

Data are presented as the mean ± SE with ANOVA *p*-value; means in columns followed by the same letter are not significantly different at the 5% level of probability (*p* < 0.05).

**Table 3 molecules-27-06979-t003:** The content of total chlorophylls and individually identified chlorophyll forms in apple (in mg/100 g FW).

	Total Chlorophylls	Chlorophyll b	Chlorophyll a	Chlorophyll a/b
Time of storage				
Control	0.40 ± 0.00 ^a^	0.33 ± 0.00 ^a^	0.07 ± 0.00 ^a^	0.20 ± 0.01 ^a^
After 7 days of storage	0.37 ± 0.00 ^b^	0.32 ± 0.00 ^b^	0.06 ± 0.00 ^b^	0.19 ± 0.00 ^a^
After 14 days of storage	0.40 ± 0.00 ^a^	0.35 ± 0.00 ^a^	0.04 ± 0.00 ^c^	0.15 ± 0.00 ^b^
After 21 days of storage	0.34 ± 0.01 ^b^	0.31 ± 0.01 ^b^	0.03 ± 0.00 ^c^	0.12 ± 0.00 ^c^
Temperature of storage				
Storage at 5 degrees	0.38 ± 0.00 ^a^	0.34 ± 0.00 ^a^	0.04 ± 0.00 ^b^	0.12 ± 0.01 ^b^
Storage at 20 degrees	0.37 ± 0.00 ^a^	0.31 ± 0.00 ^a^	0.05 ± 0.00 ^a^	0.18 ± 0.01 ^a^
*p*-value				
Time (Tim)	<0.0001	<0.0001	<0.0001	<0.0001
Temperaturę (Tem)	0.7765	0.6708	0.001	0.001
Interaction Tim × Tem	<0.0001	<0.0001	<0.0001	0.0167

Data are presented as the mean ± SE with ANOVA *p*-value; means in columns followed by the same letter are not significantly different at the 5% level of probability (*p* < 0.05).

**Table 4 molecules-27-06979-t004:** The antioxidant activity (in µM TEAC/1 g) and the content of total polyphenols and individually identified phenolic acids in apple (in mg/100 g FW).

	Antioxidant Activity	Total Polyphenols	Total Phenolic Acids	Gallic	Chlorogenic	Caffeic	*p*-Coumaric	Ferulic
Time of storage								
Control	11.80 ± 0.02 ^d^	29.55 ± 0.24 ^d^	26.12 ± 0.24 ^d^	2.63 ± 0.09 ^d^	21.53 ± 0.25 ^d^	1.20 ± 0.03 ^a^	0.54 ± 0.00 ^a^	0.22 ± 0.00 ^c^
After 7 days of storage	11.91 ± 0.04 ^c^	36.56 ± 0.77 ^c^	27.34 ± 0.52 ^c^	3.90 ± 0.12 ^c^	21.73 ± 0.39 ^c^	0.96 ± 0.01 ^b^	0.52 ± 0.01 ^b^	0.22 ± 0.00 ^c^
After 14 days of storage	13.00 ± 0.01 ^b^	73.30 ± 0.30 ^b^	56.73 ± 0.21 ^b^	19.87 ± 0.13 ^b^	35.46 ± 0.22 ^b^	0.63 ± 0.01 ^c^	0.49 ± 0.00 ^b^	0.28 ± 0.01 ^b^
After 21 days of storage	13.80 ± 0.07 ^a^	107.65 ± 3.34 ^a^	96.84 ± 3.25 ^a^	47.40 ± 1.61 ^a^	47.98 ± 1.63 ^a^	0.57 ± 0.00 ^d^	0.46 ± 0.01 ^c^	0.43 ± 0.00 ^a^
Temperature of storage								
Storage at 5 degrees	12.88 ± 0.02 ^b^	81.18 ± 0.24 ^a^	68.27 ± 0.24 ^a^	29.60 ± 0.09 ^a^	37.19 ± 0.25 ^a^	0.64 ± 0.03 ^b^	0.50 ± 0.00 ^a^	0.33 ± 0.00 ^a^
Storage at 20 degrees	12.93 ± 0.02 ^a^	63.84 ± 0.80 ^b^	52.34 ± 0.81 ^b^	17.85 ± 0.25 ^b^	32.92 ± 0.54 ^b^	0.80 ± 0.02 ^a^	0.48 ± 0.00 ^b^	0.29 ± 0.00 ^b^
*p*-value								
Time (Tim)	<0.0001	<0.0001	<0.0001	<0.0001	<0.0001	<0.0001	<0.0001	<0.0001
Temperature (Tem)	<0.0001	<0.0001	<0.0001	<0.0001	<0.0001	<0.0001	<0.0001	<0.0001
Interaction Tim × Tem	<0.0001	0.0008	0.0008	0.0004	<0.0001	0.0009	0.0003	0.0004

Data are presented as the mean ± SE with ANOVA *p*-value; means in columns followed by the same letter are not significantly different at the 5% level of probability (*p* < 0.05).

**Table 5 molecules-27-06979-t005:** The content of total flavonoids and individually identified flavonoids in apple (in mg/100 g FW).

	Total Flavonoids	Catechin	Epigallocatechin	Quercetin-3-O-Rutinoside	Luteolin	Quercetin	Kaempferol
Time of storage							
Control	3.42 ± 0.01 ^d^	2.49 ± 0.01 ^d^	0.32 ± 0.00 ^c^	0.02 ± 0.00 ^c^	0.20 ± 0.00 ^a^	0.17 ± 0.00 ^a^	0.21 ± 0.00 ^a^
After 7 days of storage	9.22 ± 0.24 ^c^	8.40 ± 0.23 ^c^	0.24 ± 0.00 ^d^	0.02 ± 0.00 ^c^	0.20 ± 0.00 ^a^	0.16 ± 0.00 ^b^	0.20 ± 0.00 ^b^
After 14 days of storage	16.58 ± 0.09 ^a^	15.11 ± 0.08 ^a^	0.44 ± 0.00 ^b^	0.44 ± 0.01 ^b^	0.21 ± 0.00 ^a^	0.17 ± 0.00 ^ab^	0.21 ± 0.00 ^ab^
After 21 days of storage	10.81 ± 0.09 ^b^	9.03 ± 0.09 ^b^	0.58 ± 0.01 ^a^	0.66 ± 0.01 ^a^	0.19 ± 0.01 ^a^	0.16 ± 0.00 ^b^	0.19 ± 0.01 ^b^
Temperature of storage							
Storage at 5 degrees	12.91 ± 0.01 ^a^	11.29 ± 0.01 ^a^	0.42 ± 0.00 ^b^	0.61 ± 0.00 ^a^	0.21 ± 0.00 ^a^	0.17 ± 0.00 ^a^	0.21 ± 0.00 ^a^
Storage at 20 degrees	11.50 ± 0.08 ^b^	10.40 ± 0.08 ^b^	0.42 ± 0.00 ^a^	0.14 ± 0.00 ^b^	0.19 ± 0.00 ^b^	0.16 ± 0.00 ^a^	0.19 ± 0.00 ^a^
*p*-value							
Time (Tim)	<0.0001	<0.0001	<0.0001	<0.0001	0.3053	0.0035	0.0026
Temperature (Tem)	0.0627	0.2200	<0.0001	<0.0001	0.004	0.3791	0.0005
Interaction Tim × Tem	<0.0001	<0.0001	<0.0001	<0.0001	0.0003	0.0002	0.0001

Data are presented as the mean ± SE with ANOVA *p*-value; means in columns followed by the same letter are not significantly different at the 5% level of probability (*p* < 0.05).

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
