# Peer review of "The Effect of Storage Conditions on the Content of Molecules in Malus domestica ‘Chopin’ cv. and Their In Vitro Antioxidant Activity"

_molecules, 2022, doi:10.3390/molecules27206979_

Round 1

Reviewer 1 Report (Previous Reviewer 1)

In the resubmitted manuscript, there are still some points that must be clarified before the article can be accepted for publication:

1.       What is the main focus of this manuscript? The Authors in the section on Introduction underline the advantages of organic farming. The presented results consider the apple analysis from the one certified organic orchard. Therefore, are the obtained results sufficient to indicate that food (in this case, apples) grown on organic farms is better? According to the reviewer, there is no proper comparison that organic farming stimulates a higher content in apples of biologically active compounds responsible for antioxidant properties. It seems important to prove whether, for example, the content of vitamin C, carotenoids and phenolic compounds changes (decreased ) slower if the apple comes from ecological work compared to traditional cultivation (considering the same species, the same storage conditions).

2.       To what period (7, 14 or 21 days) does the data presented in Tables 1-5 refer, considering the effect of the storage temperature (i.e. 5 and 20 degrees Celsius)? According to the authors (lines 225-226), “The apples 225 were stored for 7, 14 and 21 days at 5 and 20 degrees Celsius”. It suggests the set of apple samples of 7, 14, 21 for 5°C and the second set of samples of 7, 14, 21 for 20°C

3.       Line 223: should be “orchard” instead of “orchards”.

4.       Lines 228-229 à Authors wrote, “The remaining material was freeze-dried using a Labconco freeze-drier (−45 °C, 0.11 228 mBar), ground in a laboratory mill A-11, and stored at −80 °C before further analyses.” Please explain for what kind of analysis “the remaining material” was used. Does this mean that the samples related to the specific storage conditions were not analyzed immediately after 7, 14 and 21 days?

5.       There are still no chromatograms of HPLC analysis added that consider the polyphenols content in apple extracts. Please include in the manuscript the chromatograms considering the set of two chromatograms, the analysis of a mixture of standards and apple extracts. Was there any matrix effect? How was the presence of the specified analyte in the apple extracts confirmed to ensure there was no coelution effect?

6.       Why, on the chromatograms showing the mixture of standards, the retention times of some analytes differ from the chromatograms where the individual compounds were analysed. For example, the retention time of lutein in Fig 1 is 6 min, whereas, in Fig. 1A, it refers to almost 8 min)? The same situation considers the retention time of zeaxanthin, chlorophyll-a and b, as well as β-carotene and also some of the phenolic compounds. Furthermore, please add information about the concentration of the analytes in the analysed samples of standards.

7.       In Fig. 2, there should be “chlorogenic acid” instead of “chlorogenic”. The same consider “gallic.”

Author Response

Thank you for your reviews. Below are our responses to reviews.

In the resubmitted manuscript, there are still some points that must be clarified before the article can be accepted for publication:

  1. What is the main focus of this manuscript? The Authors in the section on Introduction underline the advantages of organic farming. The presented results consider the apple analysis from the one certified organic orchard. Therefore, are the obtained results sufficient to indicate that food (in this case, apples) grown on organic farms is better? According to the reviewer, there is no proper comparison that organic farming stimulates a higher content in apples of biologically active compounds responsible for antioxidant properties. It seems important to prove whether, for example, the content of vitamin C, carotenoids and phenolic compounds changes (decreased ) slower if the apple comes from ecological work compared to traditional cultivation (considering the same species, the same storage conditions).

Authors' reply: The apple were from the one certified organic orchard. In the introduction, a part describing organic farming has been added. There is a description of the principles of the organic system, because the research material used for the research came from such a system. However, the research itself does not focus on comparing organic and conventional raw materials. Therefore, such information was not included in the results and summary. Only in the introduction it is mentioned that there are scientific studies that have confirmed a higher content of biologically active compounds in organic raw materials compared to conventional raw materials.

  1. To what period (7, 14 or 21 days) does the data presented in Tables 1-5 refer, considering the effect of the storage temperature (i.e. 5 and 20 degrees Celsius)? According to the authors (lines 225-226), “The apples 225 were stored for 7, 14 and 21 days at 5 and 20 degrees Celsius”. It suggests the set of apple samples of 7, 14, 21 for 5°C and the second set of samples of 7, 14, 21 for 20°C.

Authors' reply: The apples were stored for 7, 14, 21 days at 5 ° C and the second set of samples 7, 14, 21 days at 20 ° C. However, the data in the tables show the mean values for all samples stored at 5 ° C, 20 ° C.

  1. Line 223: should be “orchard” instead of “orchards”.

Authors' reply: Has been corrected.

  1. Lines 228-229 à Authors wrote, “The remaining material was freeze-dried using a Labconco freeze-drier (−45 °C, 0.11 228 mBar), ground in a laboratory mill A-11, and stored at −80 °C before further analyses.” Please explain for what kind of analysis “the remaining material” was used. Does this mean that the samples related to the specific storage conditions were not analyzed immediately after 7, 14 and 21 days?

Authors' reply: All samples were immediately freeze-dried before analyzes.

  1. There are still no chromatograms of HPLC analysis added that consider the polyphenols content in apple extracts. Please include in the manuscript the chromatograms considering the set of two chromatograms, the analysis of a mixture of standards and apple extracts. Was there any matrix effect? How was the presence of the specified analyte in the apple extracts confirmed to ensure there was no coelution effect?

Authors' reply: Compound analysis chromatograms are included as supplementary material. However, it is no longer possible to make an overlapping image. The reading was made on the basis of the standards retention time.

  1. Why, on the chromatograms showing the mixture of standards, the retention times of some analytes differ from the chromatograms where the individual compounds were analysed. For example, the retention time of lutein in Fig 1 is 6 min, whereas, in Fig. 1A, it refers to almost 8 min)? The same situation considers the retention time of zeaxanthin, chlorophyll-a and b, as well as β-carotene and also some of the phenolic compounds. Furthermore, please add information about the concentration of the analytes in the analysed samples of standards.

Authors' reply: As the pressure increased on the column, the peaks shifted. At this point, all the peaks in the chromatogram were shifting, so we had no doubts in the identification of the compounds. At the time of validation of this method, internal standards were made, but they were for validation purposes.

  1. In Fig. 2, there should be “chlorogenic acid” instead of “chlorogenic”. The same consider “gallic.”

Authors' reply: Has been corrected.

Reviewer 2 Report (Previous Reviewer 3)

I cannot recommend acceptance of the paper. This manuscript has high similarity with Srednicka-Tober et al. published in Applied Sciences in 2020 (doi:10.3390/app10092997). Authors cannot just copy the  whole Materials and Methods from a previous publication.

Author Response

Thank you for your reviews. Below are our responses to reviews.

I cannot recommend acceptance of the paper. This manuscript has high similarity with Srednicka-Tober et al. published in Applied Sciences in 2020 (doi:10.3390/app10092997). Authors cannot just copy the  whole Materials and Methods from a previous publication.

Authors' reply: Thank you for all these comments and remarks that will help to improve our manuscript. Thanks for your suggestion. It has been corrected.

Round 2

Reviewer 2 Report (Previous Reviewer 3)

The manuscript was improved

This manuscript is a resubmission of an earlier submission. The following is a list of the peer review reports and author responses from that submission.

Round 1

Reviewer 1 Report

In the submitted manuscript Author describes the studies considering the effect of storage conditions on the content of some biologically active ingredients in the apple Malus domestica “Chopin” cv. and their antioxidant activity. Generally, the reader feels unsatisfied with the way of the results discussion. There are some points that must be clarified before the article can be accepted for publication. It should also be extended with some of the additional research:

  1. Line 42 -43: If there is an “increasing number of scientific publications” please add some more references, not only on review but the source's publications.
  2. Line 97: instead of Table 2 should be Table 1
  3. Considering the effect of time of storage what were the conditions of temperature. The same situation considers the effect of temperature - what was the time of storage, in this case? Please add these information in the part of “Materials and Methods”
  4. It would be interesting to include the considerations regarding the relationship between changes in the content of individual components (carotenoids, vitamin C, etc.) and the specificity of metabolic processes taking place in apple tissues. Why there is no effect of temperature on chlorophyll content whereas the temperature influence the content of carotenoids?
  5. Line 265: “Shimadzu equipment characterized in the previous chapter” - which chapter is this about?
  6. There are none chromatograms of HPLC analysis considering the polyphenols content? Was there any effect of sample matrix? Please compare the chromatograms of standards with analysed extracts.
  7. The chromatograms of HPLC analysis of carotenoids and chlorophyll must be included in the text of the manuscript or supplementary data, too.
  8. Why in some cases, the content of some analytes (e.g. phenolic compounds)  increased with time but for the other decreased.
  9. Were there any changes of pH in the tested samples? Are there any correlations of pH with the obtained results?
  10. The data of total polyphenol content TPC and phenolic compounds with the Folin-Ciocalteu are missing? It would be ease to compare the nutrition properties of studied analytes.
  11. The Author emphasises the importance of organic farming and the level of antioxidants. How the obtained results confirmed it? Is it possible to make some kind of comparison?
  12. How the obtained results correlate with the other studies, where other types of apples were investigated?. Is the content of determined analytes high in the studies samples of apple?
  13. Please enter the coordinates of the location of the apple orchards. Whether the orchards were owned by one farm or several?
  14. Spaces are not needed in some places in the manuscript text
  15. What about the sugar content? Were any sugar analysis run?

Although the overall concept has unique characteristics and the workflow seems logical, the described studies should be once verified and  owing to the above issues, I believe the manuscript is unsuitable for publication in its current form.

Author Response

In the submitted manuscript Author describes the studies considering the effect of storage conditions on the content of some biologically active ingredients in the apple Malus domestica “Chopin” cv. and their antioxidant activity. Generally, the reader feels unsatisfied with the way of the results discussion. There are some points that must be clarified before the article can be accepted for publication. It should also be extended with some of the additional research:

Comment 1: Line 42 -43: If there is an “increasing number of scientific publications” please add some more references, not only on review but the source's publications.

Author response: Thank you for all these comments and remarks that will help to improve our manuscript. Thanks for your suggestion. Some references have been added in publications:

  1. Patocka et al., “Malus domestica: A review on nutritional features, chemical composition, traditional and medicinal value,” Plants, vol. 9, no. 11, pp. 1–19, 2020, doi: 10.3390/plants9111408.
  2. Dai and R. J. Mumper, “Plant phenolics: Extraction, analysis and their antioxidant and anticancer properties,” Molecules, vol. 15, no. 10, pp. 7313–7352, 2010, doi: 10.3390/molecules15107313.
  3. Quideau, D. Deffieux, C. Douat-Casassus, and L. Pouységu, “Plant polyphenols: Chemical properties, biological activities, and synthesis,” Angew. Chemie - Int. Ed., vol. 50, no. 3, pp. 586–621, 2011, doi: 10.1002/anie.201000044.
  4. Navarro et al., “Polyphenolic characterization and antioxidant activity of malus domestica and prunus domestica cultivars from Costa Rica,” Foods, vol. 7, no. 2, 2018, doi: 10.3390/foods7020015.
  5. Wolfe, X. Wu, and R. H. Liu, “Antioxidant activity of apple peels,” J. Agric. Food Chem., vol. 51, no. 3, pp. 609–614, 2003, doi: 10.1021/jf020782a.
  6. Xu et al., “Variation in phenolic compounds and antioxidant activity in apple seeds of seven cultivars,” Saudi J. Biol. Sci., vol. 23, no. 3, pp. 379–388, 2016, doi: 10.1016/j.sjbs.2015.04.002.
  7. A. Hyson, “A comprehensive review of apples and apple components and their relationship to human health,” Adv. Nutr., vol. 2, no. 5, pp. 408–420, 2011, doi: 10.3945/an.111.000513.

Comment 2: Line 97: instead of Table 2 should be Table 1

Author response: I changed the table number.

Comment 3: Considering the effect of time of storage what were the conditions of temperature. The same situation considers the effect of temperature - what was the time of storage, in this case? Please add these information in the part of “Materials and Methods”

Author response: The apples were stored for 7, 14 and 21 days at 5 and 20 degrees Celsius. These information have been added in the part of “Materials and Methods”

Comment 4: “…: It would be interesting to include the considerations regarding the relationship between changes in the content of individual components (carotenoids, vitamin C, etc.) and the specificity of metabolic processes taking place in apple tissues. Why there is no effect of temperature on chlorophyll content whereas the temperature influence the content of carotenoids?..”

Author response: Thank you for pointing such interesting observation direction. Unfortunately in our experiment no metabolic processes in apple tissues were not measured. We focus our attention only on the composition and concentration of bioactive compounds. But Author taking Reviewer suggestion under taking and in time of next experiment it will be measured. Thank you again for pointing a new direction of observations and analysis.

Carotenoids are one of the most dynamic dye compounds. Temperature can change concentration of carotenoids in plant tissues (apple flesh and skin). Concentration of carotenoids in our experiment is changing according to temperatures and time, but chlorophylls no. Probably chlorophylls are more chemically stables compare to carotenoids. On the other hand if we want to observe changing of chlorophylls concentration changings the products of chlorophylls degradation - chlorophyllides it should be measured. In our experiment we don’t have such possibility.

Comment 5:    Line 265: “Shimadzu equipment characterized in the previous chapter” - which chapter is this about?

Author response: I uded using Shimadzu HPLC-set (Shim-pol, Warsaw, Poland). This information have been added in the part of “Materials and Methods”.

Comment 6:     There are none chromatograms of HPLC analysis considering the polyphenols content? Was there any effect of sample matrix? Please compare the chromatograms of standards with analysed extracts.

 The chromatograms of HPLC analysis of carotenoids and chlorophyll must be included in the text of the manuscript or supplementary data, too.

Author response: According to Reviewer suggestion chromatogram picture both polyphenols and carotenoids in Chopin (unmatured or green) were added to manuscript text. According to used methodology only external standards were used for bioactive compounds identification and concentration calculation. No internal standards were used. 

Comment 7: Why in some cases, the content of some analytes (e.g. phenolic compounds)  increased with time but for the other decreased.

Author response:  According to metabolic synthesis pathway some phenolics are both substrates and products of synthesis reaction. A very detailed measurement including enzym compounds is required to explain the order of appearance and changes within individual polyphenolic compounds. Polyphenols are a very dynamic group of bioactive compounds and in our experiment it was not possible to measure all this pathway. Of course it is a very interesting point and probably in the future Autor focus their attention as well try to explain such phenomena. In recent literature there is no explanation as well.

Comment 8: Were there any changes of pH in the tested samples? Are there any correlations of pH with the obtained results?

Author response:  Unfortunately, the pH of the tested fruit samples was not measured. In the next planned experiment this measurement will be undertaken.

Comment 9:     The data of total polyphenol content TPC and phenolic compounds with the Folin-Ciocalteu are missing? It would be ease to compare the nutrition properties of studied analytes.

Author response: Thank you for substantial comment, but no TPC by Folin-Ciocalteu was measured. Only calculation (sum of identified individual phenolic compounds) was presented in tables as a total polyphenols.

Comment 10: The Author emphasises the importance of organic farming and the level of antioxidants. How the obtained results confirmed it? Is it possible to make some kind of comparison?

Author response: The organic system has a positive effect on the content of polyphenols in plants. Since this system does not use synthetic pesticides, the plants need to build up their natural defense mechanism. This mechanism consists of polyphenolic compounds called natural pesticides. The comparison could be made by comparing the tested organic apples with their conventional counterparts.

Comment 11: How the obtained results correlate with the other studies, where other types of apples were investigated?. Is the content of determined analytes high in the studies samples of apple?

Author response: The comparison of the obtained results with the results of other researchers was made in the Discussion chapter. These results are comparable, and the content of antioxadants in the tested samples was high for this type of fruit.

Comment 12: Please enter the coordinates of the location of the apple orchards. Whether the orchards were owned by one farm or several?

Author response:  Fruits of the apple cultivar (Chopin) were harvested in the 2020 vegetation season from certified organic orchards located in central Poland (Łódź region, Marchaty (51°47′8″N 20°29′13″E)). This information have been added in the part of “Materials and Methods”.

Comment 13: Spaces are not needed in some places in the manuscript text

Author response: Thank you. I corrected it.

Comment 14:  What about the sugar content? Were any sugar analysis run?

Author response: Unfortunately, the sugar content of the tested fruit samples was not measured. In the next planned experiment this measurement will be undertaken.

Although the overall concept has unique characteristics and the workflow seems logical, the described studies should be once verified and owing to the above issues, I believe the manuscript is unsuitable for publication in its current form.

Reviewer 2 Report

The following are minor concerns.

1)    In their study authors reported that vitamin-C activity decreased as the storage time was increased. However, in the conclusion part, they said antioxidant activity increased upon storage. Vitamin C is also an antioxidant. So, they should modify their conclusion to reflect this fact

2)    Similarly, carotenoids also decreased upon storage and authors should give some explanation or possible mechanism for their decrease upon storage.

3)    The authors reported an increase in polyphenols and flavonoids upon storage. One of the possibilities could be degradation or secondary metabolite formation upon storage. The authors should clarify this point.

4)    In their efforts to report the effect of temperature on the quality of the foods, they mentioned only one point. It raises the question of whether they did the experiments at 7, 14 and 21 days also at 50C and 200C or just only a one-time point.

Author Response

The following are minor concerns.

Comment 1: In their study authors reported that vitamin-C activity decreased as the storage time was increased. However, in the conclusion part, they said antioxidant activity increased upon storage. Vitamin C is also an antioxidant. So, they should modify their conclusion to reflect this fact

Author response: Thank you for all these comments and remarks that will help to improve our manuscript. Thanks for your suggestion. The final conclusions were modified taking into account the reviewer's suggestion.

‘The samples of apples stored the longest, i.e., for 21 days, had significantly the highest antioxidant activity. The main antioxidants present in apples are vitamin C and poly-phenols. Although the antioxidant activity of apples increased with the storage period, the content of vitamin C decreased with time. Therefore, it seems that the increasing antioxidant activity was positively correlated with the increasing polyphenol content in the stored apples, because the samples of apples stored the longest, i.e., for 21 days, had significantly the highest content of total polyphenol, total phenolic acid, gallic and chlorogenic contents.’

Comment 2: Similarly, carotenoids also decreased upon storage and authors should give some explanation or possible mechanism for their decrease upon storage.

Author response: This suggestion was also included in the final conclusions.

‘Carotenoids were degraded during the storage of the apples. During storage, oxidation reactions take place, in particular of vitamin C and carotenoids. As it grows storage time and temperature are decreasing the share of the yellow color and clarity the product as carotenoids are degraded.’

Thank you for pointing such interesting observation direction. Unfortunately in our experiment no metabolic processes in apple tissues were not measured. We focus our attention only on the composition and concentration of bioactive compounds. But Author taking Reviewer suggestion under taking and in time of next experiment it will be measured. Thank you again for pointing a new direction of observations and analysis.

Carotenoids are one of the most dynamic dye compounds. Temperature can change concentration of carotenoids in plant tissues (apple flesh and skin). Concentration of carotenoids in our experiment is changing according to temperatures and time, but chlorophylls no. Probably chlorophylls are more chemically stables compare to carotenoids. On the other hand if we want to observe changing of chlorophylls concentration changings the products of chlorophylls degradation - chlorophyllides it should be measured. In our experiment we don’t have such possibility.

Comment 3: The authors reported an increase in polyphenols and flavonoids upon storage. One of the possibilities could be degradation or secondary metabolite formation upon storage. The authors should clarify this point.

Author response: I agree with the reviewer. I clarify this point in the final conclusions.

‘An important finding is that an increase in the content of polyphenols and flavonoids has been noticed during storage. It turns out that the storage conditions caused a reaction which resulted in the production of secondary metabolites, which are polyphenolic compounds. Polyphenols are the natural protective mechanism of the plant, so the plant synthesizes higher amounts of polyphenols to protect itself against external factors.’

According to metabolic synthesis pathway some phenolics are both substrates and products of synthesis reaction. A very detailed measurement including enzyme compounds is required to explain the order of appearance and changes within individual polyphenolic compounds. Polyphenols are a very dynamic group of bioactive compounds and in our experiment it was not possible to measure all this pathway. Of course it is a very interesting point and probably in the future Autor focus their attention as well try to explain such phenomena. In recent literature there is no explanation as well.

Comment 4: In their efforts to report the effect of temperature on the quality of the foods, they mentioned only one point. It raises the question of whether they did the experiments at 7, 14 and 21 days also at 50C and 200C or just only a one-time point.

Author response: At each of the tested times, i.e. after 7, 14 and 21 days, the apples were stored in two temperature ranges 5 and 20 degrees.

Reviewer 3 Report

I cannot recommend acceptance of the paper in the present state. This might be possible only after the authors submit a completely rewritten version. This manuscript has high similarity with Srednicka-Tober et al. published in Applied Sciences in 2020 (doi:10.3390/app10092997).

Author Response

Comment 1: I cannot recommend acceptance of the paper in the present state. This might be possible only after the authors submit a completely rewritten version. This manuscript has high similarity with Srednicka-Tober et al. published in Applied Sciences in 2020 (doi:10.3390/app10092997).

Author response: Thank you for all these comments and remarks that will help to improve our manuscript. As suggested by the author, similarities with the publication of Średnicka-Tober et al. published in Applied Sciences in 2020 (doi: 10.3390 / app10092997), which I am co-author of, have been significantly changed. However, this work has a completely different content. In the previous study, we aimed to compare the polyphenol content in apples of different cultivars from the organic and conventional systems, while in this study I investigated the effect of different storage conditions for apples of the Chopin cultivars.

Round 2

Reviewer 1 Report

Manuscript ID: molecules-1737074

In the revised manuscript there are still some points that must be clarified before the article can be accepted for publication:

1.       Line 12: should be “alternative” insted of „al-ternative”

2.       Lines 17-19: Author indicated that „the aim of this study was to  evaluate and compare the content of a number of bioactive compounds and the antioxidant activity  of fruits of the local apple ‘Chopin’ cultivar grown in certified organic orchards in Poland” – how many orchards did the apples come from? In the experimental section only one location was given.

3.       There are still none chromatograms of HPLC analysis considering the polyphenols content added. Please compare the chromatograms considering both: i. vitamin C, ii. Carotenoids and chlorophyls as well iii.phenolic compounds as the sets of two chromatograms of standards analysis and analyse extracts.

4.       The Fig 1 and Fig 2 should be corrected as indicated above. Furthermore, in Fig 1, the information which peak at the chromatogram (indicate as number)  corresponds to which analytes should be added. What was the concentration of the analytes in the analysed samples. Was it sample of standards or extracts? Please take care of the details, for example: chlorogenic acid instead of  chlorogenic.

4. Line 72: after word „not” the space is not necessery.

5. Fig 2, how Author confirmed that the peaks indicated as 1, 3, 4, 5 are really the compounds from the goup of the phenols?

6.  Please indicate the type of detector used for Vitamin C analysis.

Reviewer 3 Report

There is a still a high similarity with previously reported manuscript as well as with: Vondráková et al. The effect of storage conditions on the carotenoid and phenolic acid contents of selected apple cultivars. Eur Food Res Technol 246, 1783–1794 (2020). https://doi.org/10.1007/s00217-020-03532-w and 

Please rewrite the lines 48-84, 168-177, 194-207, 211-217, 231-239, all Material and Methods Section: you cannot simply copy the text from the previously published manuscript.